# An analysis of NHS 111 demand for primary care services: A retrospective cohort study

**Richard Pilbery**[1,2]☯*, **Madeleine Smith**[3]☯, **Jonathan Green**[4]☯, **Daniel Chalk**[5]☯, **Colin O'Keeffe**[2]☯

**1** Yorkshire Ambulance Service Research Institute, Yorkshire Ambulance Service NHS Trust, Wakefield, West Yorkshire, United Kingdom, **2** Centre for Urgent and Emergency Care Research (CURE), Health Services Research Section, University of Sheffield, Sheffield, South Yorkshire, United Kingdom, **3** Business Intelligence, NHS Devon, Exeter, Devon, United Kingdom, **4** School of Health Professions, Faculty of Health, University of Plymouth, Plymouth, United Kingdom, **5** University of Exeter Medical School, University of Exeter, Exeter, United Kingdom

☯ These authors contributed equally to this work.
* r.s.pilbery@sheffield.ac.uk

**Data Availability Statement:** Data cannot be shared publicly because the study dataset was derived from Connected Yorkshire research database data, which has strict controls on access

## Abstract

The NHS 111 service triages over 16,650,745 calls per year and approximately 48% of callers are triaged to a primary care disposition, such as a telephone appointment with a general practitioner (GP). However, there has been little assessment of the ability of primary care services to meet this demand. If a timely service cannot be provided to patients, it could result in patients calling 999 or attending emergency departments (ED) instead. This study aimed to explore the patient journey for callers who were triaged to a primary care disposition, and the ability of primary care services to meet this demand. We obtained routine, retrospective data from the Connected Yorkshire research database, and identified all 111 calls between the 1st January 2021 and 31st December 2021 for callers registered with a GP in the Bradford or Airedale region of West Yorkshire, who were triaged to a primary care disposition. Subsequent healthcare system access (111, 999, primary and secondary care) in the 72 hours following the index 111 call was identified, and a descriptive analysis of the healthcare trajectory of patients was undertaken. There were 56,102 index 111 calls, and a primary care service was the first interaction in 26,690/56,102 (47.6%) of cases, with 15,470/26,690 (58%) commenced within the specified triage time frame. Calls to 999 were higher in the cohort who had no prior contact with primary care (58% vs 42%) as were ED attendances (58.2% vs 41.8), although the proportion of avoidable ED attendances was similar (10.5% vs 11.8%). Less than half of 111 callers triaged to a primary care disposition make contact with a primary care service, and even when they do, call triage time frames are frequently not met, suggesting that current primary care provision cannot meet the demand from 111.

## Introduction

The National Health Service (NHS) 111 service aims to assist members of the public with urgent medical care needs and is the successor to the NHS Direct service in England.

as part of the ethical approvals in place for the database. However, the data are available for researchers who meet the criteria for access to confidential data, by making an application to the Bradford Institute for Health Research (contact via email: bradfordresearch@bthft.nhs.uk).

**Funding:** This report is independent research funded by the National Institute for Health and Care Research, Yorkshire and Humber (reference NIHR200166, RP, CO) and South West Pennisula (reference NIHR200167, DC) Applied Research Collaborations. The funders had no role in study design, data collection and analysis, decision to publish, or preparation of the manuscript.

**Competing interests:** The authors have declared that no competing interests exist.

Following pilots in four sites it was rolled out nationally, with the final site going live in England in 2014, and in 2019/20 111 received over 19 million calls [1]. Its key founding objective was to provide easy access to support for the public with urgent care needs, to ensure they received the "right care, from the right person, in the right place, at the right time" [2]. It is also the key component of the 24/7 Integrated Urgent Care Service outlined in the NHS Long Term Plan [3].

The proposed benefits of this system were to improve the public's access to urgent healthcare, help people use the right service first time including self-care and provide commissioners with management information regarding the usage of services.

Initial evaluation of the four pilot sites reported that the public were generally satisfied with the service and followed the advice given, there were no significant impacts on emergency department or urgent care service utilisation, but there were increases in 999 ambulance service activity as a result of the introduction of the 111 service [4, 5].

Subsequent evaluation of the service has explored the effect of clinical input on triage decisions with respect to patient compliance and avoidable emergency department attendance [6–8]. However, no studies have been conducted using data collected following the publication of the Integrated Urgent Care Specification, published in 2017, which called for sufficient numbers of clinicians, working to approved guidelines and protocols, to support 111 call handlers [9]. In addition, there has been little scrutiny of the ability of primary care provision (particularly out-of-hours) to meet the demand of the NHS 111 service. This is particularly pertinent, since approximately 55% of all NHS 111 call dispositions result in a referral to a primary care service. If a timely service cannot be provided to patients, it is possible that this will result in patients calling 999 or attending emergency departments (ED) directly.

The aim of this study was to explore the patient journey for callers who are given a primary care disposition following a call to NHS 111, and the ability of primary care services to meet relevant 111 call dispositions. The primary objective was to determine the proportion of initial healthcare contacts following the index 111 call, that were a primary care service. Secondary objectives included determining what proportion of primary care service contacts were made within the specified triage timeframe, and for ED admissions, the proportion of attendances that were avoidable, stratified by tine of attendance and whether an initial contact had been made with a primary care service or not.

## Methods

### 111 call triage and disposition

The 111 service uses the Clinical Decision Support System (CDSS) NHS Pathways to triage calls. It is not intended to be a diagnostic system, but instead is designed to assess symptoms and signpost to onward care, if required. Calls handlers are non-clinical, but work with clinicians who can provide support and, in some circumstances, take over the call [10].

NHS Pathways comprises an interlinked series of algorithms (pathways) that link questions and care advice resulting in a clinical endpoint known as a disposition. This specifies the general category of service and the time frame that this should be available to the caller. These pathways correspond to a symptom group (SG), such as chest pain or headache, and a symptom discriminator, which describes the level of care required. Triage questioning continues until a relevant symptom related to a condition cannot be safely excluded and the patient is allocated a symptom discriminator which describes the appropriate level of care required, for example 'full Primary Care assessment and prescribing capability' [11].

## Data

We obtained routine, retrospective data from the Connected Yorkshire research database, which provides linked data for approximately 1.2 million citizens across the Bradford and Airedale region of Yorkshire [12]. Datasets include 111 and 999 call data, as well as primary and secondary care (including emergency department and in-patient activity). All datasets are pseudonymised so that researchers cannot identify individual participants.

We obtained a convenience sample of all 111 calls between the 1st January 2021 and 31st December 2021 for patients who were triaged to a primary care disposition (S1 Table) and registered with a General Practitioner (GP) in the Bradford area at the time of the call. Depending on perceived acuity as determined by the NHS Pathways system, patients are allocated to either a face-to-face or telephone consultation with a primary care clinician within a specified time frame. Subsequent healthcare system access in the following 72 hours following the first (index) call was identified, by searching the 111 and 999 call, primary care, and hospital emergency department and in-patient admission datasets.

## Analysis

We conducted a descriptive analysis comparing patient demographic, triage characteristic and patient trajectory data for patients who did, and did not, receive a timely contact with a primary care service. The primary outcome measure was the proportion of initial healthcare contacts that were made to a primary care service following the index 111 call, and reported as counts and percentages. The first secondary outcome measure was the proportion of primary care service contacts that were made within the time specified by index 111 call triage. As before, counts and percentages were reported. For the secondary outcome measure determining the proportion of ED admissions that were classed as avoidable, we calculated counts and percentages of attendances that met criteria for avoidable admission as defined by O'Keeffe et al [13]. They defined an avoidable attendance as a patient presenting to a consultant-led ED which provides a 24-hour service with full resuscitation facilities and designated accommodation for the reception of emergency care patients (referred to as a type 1 ED [14]), but who do not receive investigations, treatments or referral that required the facilities of that ED. The results were stratified by whether the attendance was 'in-hours' (between 08:00 and 18:00 on a weekday) and if a primary care service had been contacted prior to ED attendance.

To visualise the patient's trajectories, we generated a sankey diagram. All analysis was conducted using the statistics package, R [15].

## Ethical approval

This study was approved by the Bradford Learning Health System Board in accordance with the Connected Yorkshire NHS Research Ethics Committee (REC) approval relating to the Connected Yorkshire research database (17/EM/0254). No separate Health Research Authority (HRA) approval was required for this study.

## PPI

The application and protocol for this study was review by the Yorkshire Ambulance Service NHS Trust patient research ambassador. In addition, Connected Bradford have an active patient and public involvement group who were involved in the decision to approve this study.

## Results

Between the 1st January 2021 and 31st December 2021, there were 56,102 index 111 calls with a primary care disposition. The first healthcare interaction following the call was a primary care service in 26,690/56,102 (47.6%) of cases. However, in 21,749/56,102 (38.8%) of cases, the caller had no further healthcare contact in the 72 hours following the index 111 call (Table 1).

During the week, calls were most commonly made after 18:00, consistent with coinciding with a working-age demographic finishing a 'typical' working day, whereas calls were spread more widely across the day at the weekend (Fig 1). There were 190 distinct symptom groups in the data, although the most common were pain and/or frequency when passing urine, unwell infants and rashes (Table 1 and S1 Fig). The median age of callers was 29 years (IQR 8–50 years), although the distribution of ages was bimodal, with peaks seen in patients less than a year old, and in patients aged between 20–30 years (S2 Fig). Callers were more commonly female across virtually the entire age range.

### Referral services and clinical advisor involvement in call handling

While all included cases received a triage disposition of contact with a primary care service, services in this category do not only include GPs and integrated urgent care (IUC) centres. Pharmacists, opticians and maternity, mental health and community-based services are also included. In this cohort, 'alternatives' to GP or IUC services were frequently rejected for a variety of reasons including patient preference and service-based constraints, such as capacity issues (Table 2). Only GP appointments appeared to be bookable by the 111 call handler based on the data in this cohort, although this was infrequently undertaken and mostly 'in-hours' (S2 Table).

Greater emphasis has been placed on the availability of skilled clinicians to support the non-clinical call handlers [9]. However, in patients with a primary care service disposition, clinicians infrequently take over calls, irrespective of triage acuity (S3 Table). However, it is possible that clinical advice is provided to call handlers without the clinician actually taking over the call themselves, which would not appear in our data.

### Patient healthcare trajectory

In most cases, patients either had contact with a primary care service and no further healthcare interaction, or did not have contact with a healthcare service at all (41,529/56,102, 74%) (Fig 2). However, despite the short follow-up (72 hours), there were 1,091/56,102 (1.9%) of patients who received more than 5 healthcare interactions in that period.

### GP contacts

Following the index call, the first healthcare service contact was with a primary care service in 26,690 of callers (Table 3) Perhaps unsurprisingly, triage contact times of one hour were the most challenging to meet with only 2,273/6,100 (37%) occurring within the specified triage time frame, despite representing callers triaged to the highest acuity. There was a higher proportion of callers who visited an ED following contact with a primary care service within the time frame (1,442/2,311, 62%), although it is unclear from the data why this should be the case.

### Emergency department attendance

There were 9,290 emergency department attendances and 1,029 (11.1%) met the [13] definition of an avoidable attendance. In summary, a patient is defined as meeting this definition when they present to a consultant-led ED which provides a 24-hour service with full

**Table 1. Summary data for index 111 calls with a primary care disposition.**

| Characteristic | Primary care first contact, N = 26,690 | Other healthcare service first contact, N = 7,663 | No healthcare contact in 72 hours, N = 21,749 | Overall, N = 56,102 |
|---|---|---|---|---|
| **Triaged primary care contact timeframe (N, %)** | | | | |
| 1hr | 6,100 (23%) | 1,695 (22%) | 2,553 (12%) | 10,348 |
| 2hrs | 9,966 (37%) | 2,893 (38%) | 6,921 (32%) | 19,780 |
| 6hrs | 6,137 (23%) | 1,570 (20%) | 5,187 (24%) | 12,894 |
| >6hrs | 4,487 (17%) | 1,505 (20%) | 7,088 (33%) | 13,080 |
| **Patient age in years (median, IQR)** | 28 (5, 50) | 30 (13, 51) | 30 (15, 49) | 29 (8-50) |
| **Patient sex (N, %)** | | | | |
| Female | 15,978 (60%) | 4,618 (60%) | 13,462 (62%) | 34,058 |
| Male | 10,711 (40%) | 3,045 (40%) | 8,286 (38%) | 22,042 |
| Unknown | 1 (<0.1%) | 0 (0%) | 1 (<0.1%) | 2 |
| **Time of index 111 call (N, %)** | | | | |
| Out-of-hours | 21,314 (80%) | 5,656 (74%) | 14,360 (66%) | 41,330 |
| In-hours | 5,376 (20%) | 2,007 (26%) | 7,389 (34%) | 14,772 |
| **Primary care consultation type (N, %)** | | | | |
| Face to face | 17,879 (67%) | 5,219 (68%) | 16,603 (76%) | 39,701 |
| Telephone | 8,811 (33%) | 2,444 (32%) | 5,146 (24%) | 16,401 |
| **Primary care appointment made by 111 (N, %)** | | | | |
| No | 24,862 (93%) | 7,020 (92%) | 18,181 (84%) | 50,063 |
| Yes | 1,828 (6.8%) | 643 (8.4%) | 3,568 (16%) | 6,039 |
| **Clinical advisor involved in call (N, %)** | | | | |
| No | 22,178 (83%) | 6,352 (83%) | 17,749 (82%) | 46,279 |
| Yes | 4,512 (17%) | 1,311 (17%) | 4,000 (18%) | 9,823 |
| **Initial disposition service rejected (N, %)** | | | | |
| No | 23,036 (86%) | 6,549 (85%) | 17,344 (80%) | 46,929 |
| Yes | 3,654 (14%) | 1,114 (15%) | 4,405 (20%) | 9,173 |
| **Triage symptom group (N, %)** | | | | |
| Other | 15,392 (58%) | 4,822 (63%) | 13,197 (61%) | 33,411 |
| Pain and/or Frequency Passing Urine | 1,622 (6.1%) | 292 (3.8%) | 1,340 (6.2%) | 3,254 |
| Unwell, Under 1 Year Old | 1,291 (4.8%) | 364 (4.8%) | 840 (3.9%) | 2,495 |
| Skin, Rash | 1,205 (4.5%) | 247 (3.2%) | 988 (4.5%) | 2,440 |
| Earache | 1,313 (4.9%) | 174 (2.3%) | 915 (4.2%) | 2,402 |
| Sore Throat or Hoarse Voice | 1,174 (4.4%) | 273 (3.6%) | 871 (4.0%) | 2,318 |
| Chest and Upper Back Pain | 984 (3.7%) | 331 (4.3%) | 793 (3.6%) | 2,108 |
| Vomiting | 1,046 (3.9%) | 366 (4.8%) | 644 (3.0%) | 2,056 |
| Lower Back Pain | 895 (3.4%) | 248 (3.2%) | 805 (3.7%) | 1,948 |
| Cough | 859 (3.2%) | 244 (3.2%) | 762 (3.5%) | 1,865 |
| Abdominal Pain | 898 (3.4%) | 301 (3.9%) | 590 (2.7%) | 1,789 |
| Unknown | 11 | 1 | 4 | 16 |
| **First service contacted following index 111 call (N, %)** | | | | |
| GP | 26,690 (100%) | 0 (0%) | 0 (0%) | 26,690 |
| No further healthcare contact | 0 (0%) | 0 (0%) | 21,749 (100%) | 21,749 |
| ED | 0 (0%) | 3,803 (50%) | 0 (0%) | 3,803 |
| IUC | 0 (0%) | 2,602 (34%) | 0 (0%) | 2,602 |
| 999 | 0 (0%) | 739 (9.6%) | 0 (0%) | 739 |
| IP | 0 (0%) | 519 (6.8%) | 0 (0%) | 519 |

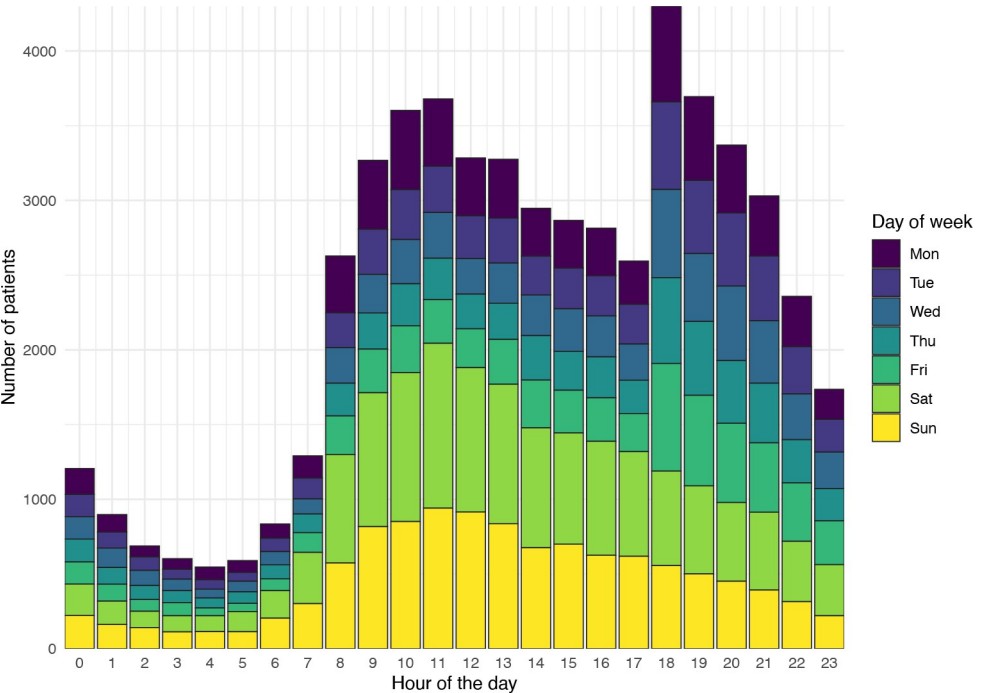

**Fig 1. 111 call volume by hour and day of week.**

resuscitation facilities and designated accommodation for the reception of emergency care patients (referred to as a type 1 ED [15]), but do not receive investigations, treatments or referral that requires the facilities of a type 1 ED.

The proportion of avoidable attendances was higher in cases where the patient had contacted a primary care service after the index 111 call (Table 4). Patients who had not previously contacted a primary care service prior attended sooner than those who had, and this trend was more pronounced out-of-hours.

## Discussion

In our study, just under half (47.6%) of callers to 111 who were triaged to a primary care service disposition contacted a primary care service as their first post-call healthcare interaction. In addition, triaged time frames of 2 hours or less were frequently not met even when contact with a primary care service was made, suggesting primary care services are struggling to meet the demand from 111. However, despite this, the rate of contact with primary care services was higher in this study than has been reported elsewhere. For example [16], linked 111 call data

**Table 2. Healthcare services referred to or rejected following 111 call triage.**

| Service category | Service accepted | Service rejected | Total Services Offered | Proportion rejected (%) |
|---|---|---|---|---|
| IUC/GP | 54,016 | 2,854 | 56,870 | 5.0 |
| Pharmacy | 1,145 | 4,073 | 5,218 | 78.1 |
| Community service | 355 | 1,976 | 2,331 | 84.8 |
| Mental health service | 40 | 119 | 159 | 74.8 |
| Optician | 18 | 117 | 135 | 86.7 |
| Maternity service | 11 | 29 | 40 | 72.5 |

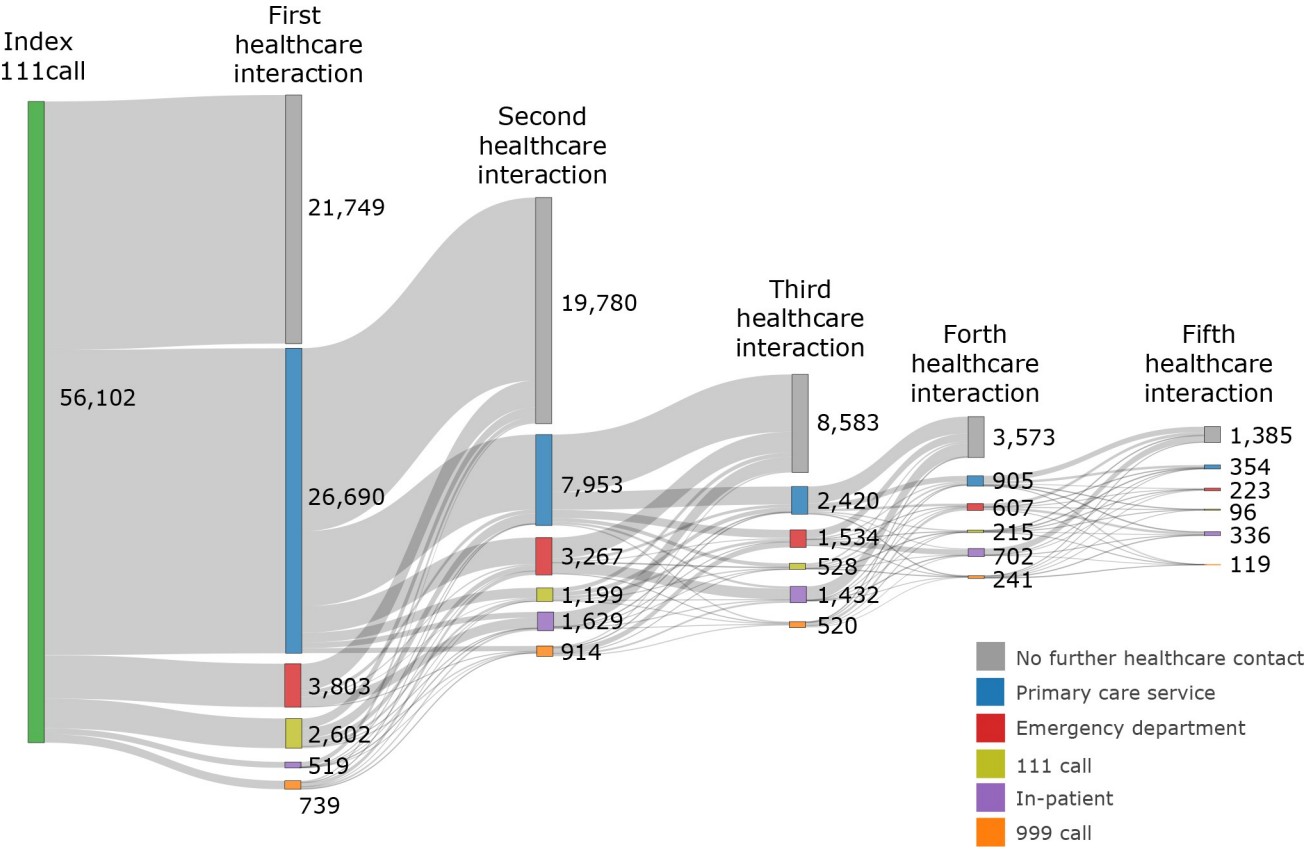

**Fig 2. Sankey diagram of healthcare service access by patients following index 111 call.**

**Table 3. Summary data for primary care contacts following index 111 call.**

|  | Primary care contact within triage timeframe | |  |
| --- | --- | --- | --- |
| Characteristic | no, N = 11,220 | yes, N = 15,470 | Overall, N = 26,690 |
| **Time of index 111 call (N, %)** |  |  |  |
| In-hours | 1,927 (17%) | 3,449 (22%) | 5,376 (20%) |
| Out-of-hours | 9,293 (83%) | 12,021 (78%) | 21,314 (80%) |
| **Triaged primary care contact timeframe (N, %)** |  |  |  |
| 1hr | 3,827 (34%) | 2,273 (15%) | 6,100 (23%) |
| 2hrs | 4,840 (43%) | 5,126 (33%) | 9,966 (37%) |
| 6hrs | 1,806 (16%) | 4,331 (28%) | 6,137 (23%) |
| >6hrs | 747 (6.7%) | 3,740 (24%) | 4,487 (17%) |
| **Next service following primary care contact (N, %)** |  |  |  |
| Ambulance service | 210 (1.9%) | 237 (1.5%) | 447 (1.7%) |
| Emergency department | 869 (7.7%) | 1,442 (9.3%) | 2,311 (8.7%) |
| In-patient | 183 (1.6%) | 306 (2.0%) | 489 (1.8%) |
| No further healthcare contact in 72 hours | 6,582 (59%) | 9,438 (61%) | 16,020 (60%) |
| Primary care | 3,022 (27%) | 3,537 (23%) | 6,559 (25%) |
| Subsequent 111 call | 354 (3.2%) | 510 (3.3%) | 864 (3.2%) |

**Table 4. Summary data for first ED attendance following index 111 call.**

| Time of attendance | Primary care service contacted prior to attendance | Avoidable attendance | Total attendances | Proportion of avoidable attendances | Median time from index call to ED attendance (hrs, IQR) |
|---|---|---|---|---|---|
| In-hours | Yes | 115 | 1,105 | 10.4 | 4.4 (2.3–20) |
| In-hours | No | 121 | 1,457 | 8.3 | 3.5 (1.6–17) |
| Out-of-hours | Yes | 345 | 2,778 | 12.4 | 7 (3.4–21.2) |
| Out-of-hours | No | 448 | 3,950 | 11.3 | 4 (1.8–15) |

with primary and secondary services in London between 2013–2017 and reported only 35% of callers triaged to a primary care disposition had contact with a GP. In contrast, experimental statistics from NHS Digital suggest that patients in the Bradford area were less likely to attend a planned GP appointment than elsewhere in England in 2021. Did-not-attend (DNA) rates for Bradford at that time were 24.7% (35.3% if cases where an appointment attendance was unknown are excluded) compared to an English mean of 8.6% [17]. Direct booking of a primary care service by 111 call handlers was associated with a higher proportion of no further healthcare system contacts, although numbers were relatively small and bookable appointments being limited mostly to in-hours consultations with a GP. Clinical advisors were involved in approximately 17.5% of all calls, although there appeared to be little to differentiate calls which did, or did not, have a clinician involved.

A systematic review by [18] identified several reasons why patients do not attend GP appointments, including work or family/childcare commitments, transport issues (including weather-related) and demographic factors such as younger age, female sex and low socio-economic background, which are disproportionally represented in our data. In addition, over 70% of planned contacts with a primary care service were face-to-face, during the third English lock down for COVID-19, and some patients may have been reluctant to attend.

While this might have resulted in the easing of the workload of primary care (and other healthcare) services, it does raise the concern that callers are not having their healthcare needs met. For example, during 2021 the incidence per patient of cardiovascular conditions such as atrial fibrillation, congestive heart disease and stroke remained below pre-pandemic levels, suggesting new diagnoses had not been made (and therefore treatment not commenced) with potential implications for patient morbidity [19].

Where contact was made with another service after the index call, this was most commonly presentation at an ED, which occurred in around 7% of cases and is similar to other studies using linked data [16, 20]. Over 10% of these attendances were classed as non-urgent, i.e. an avoidable attendance; a similar rate to those who had made contact with a primary care service before attending an ED. The reasons for this are not clear in our data, but have been explored elsewhere, and include risk minimisation by patients and carers, perceived need for a prompt healthcare intervention, compliance with instructions from healthcare professionals (in the case of those who did speak to a primary care service) and a perception that care provided by an ED is superior to alternatives [21].

## Strengths and weaknesses

To our knowledge, this study represents the most up-to-date analysis of the 111 service. Previous studies utilising linked data to undertake analysis of caller trajectories following a 111 call are dated, using data from 2017 or earlier. However, the provision of urgent and emergency care remains challenging, due in part to the COVID-19 pandemic [22] and the data presented here was collected during the third English lock down. As such, caller behaviours and presentations might be different if the study was repeated now.

While the Connected Yorkshire research database has great utility for researchers wishing to explore how patients traverse the wider healthcare system, it is restricted to a discrete geographical region in West Yorkshire, which may affect the generalisability of the results we have reported. Bradford is mainly a urban area and the 13th most deprived local authority in England (out of 333) based on the Index of Multiple Deprivation [23].

Primary care disposition includes services in addition to GP and IUC centres, meaning that interactions between a caller and healthcare service provided, for example a pharmacist, would not have been captured in the data. This means that there will be gaps in our understanding of patient journeys post-call. However, given the high proportion of alternative services which were rejected by patients in our data, this may not be a significant issue.

Finally, the reasons why many patients did not adhere to their allocated 111 dispositions can only be surmised from this data. While the study had assistance from a PPI group, this was not extended to the analysis due to lack of funding, which could have provided useful insights how patient decision making contributed to the results we have observed.

## Conclusion

Less than half of 111 callers triaged to a primary care disposition make contact with a primary care service, and even when they do, call triage time frames are frequently not met, suggesting that current primary care provision cannot meet the demand from 111.

## Supporting information

**S1 Checklist. The RECORD statement – checklist of items, extended from the STROBE statement, that should be reported in observational studies using routinely collected health data.**
(DOCX)

**S1 Table. NHS pathways primary care dispositions.**
(PDF)

**S2 Table. Direct booking by 111 call handler.**
(PDF)

**S3 Table. Clinical advisor involvement in 111 calls.**
(PDF)

**S1 Fig. Top 12 weekly 111 symptom groups allocated to callers.** The study data collection period (January to December, 2021) coincided with the third English lockdown due to COVID-19. While several symptom group weekly frequencies did not change, for example pain on passing urine, others, particularly those which might be exacerbated by the relaxing of COVID-19 restrictions, for example coughs and sore throats, did see an increase.
(TIF)

**S2 Fig. Population pyramid for index 111 calls.**
(TIF)

## Acknowledgments

This work uses data provided by patients and collected by the NHS as part of their care and support. The authors would like to express their thanks for the support provided by the team at Connected Yorkshire, especially Kuldeep Sohal and John Birkinshaw.

The views expressed in this publication are those of the author(s) and not necessarily those of the NHS, the National Institute for Health Research or the Department of Health and Social Care. The funders had no role in study design, data collection and analysis, decision to publish, or preparation of the manuscript.

## Author Contributions

**Conceptualization:** Richard Pilbery, Daniel Chalk.

**Data curation:** Richard Pilbery, Madeleine Smith.

**Formal analysis:** Richard Pilbery.

**Funding acquisition:** Colin O'Keeffe.

**Investigation:** Jonathan Green.

**Methodology:** Richard Pilbery, Daniel Chalk.

**Project administration:** Richard Pilbery, Colin O'Keeffe.

**Supervision:** Daniel Chalk.

**Writing – original draft:** Richard Pilbery, Madeleine Smith, Jonathan Green, Daniel Chalk.

**Writing – review & editing:** Richard Pilbery, Madeleine Smith, Jonathan Green, Daniel Chalk, Colin O'Keeffe.

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
