## [Decision Letter · Decision Letter 0]

9 Oct 2023

PONE-D-23-09286

An analysis of NHS 111 demand for primary care services: A retrospective cohort study

PLOS ONE

Dear Dr. Pilbery,

Thank you for submitting your manuscript to PLOS ONE. After careful consideration, we have decided that your manuscript does not meet our criteria for publication and must therefore be rejected.

I am sorry that we cannot be more positive on this occasion, but hope that you appreciate the reasons for this decision.

Kind regards,

André Ramalho, PhD

Academic Editor

PLOS ONE

Additional Editor Comments:

Our decision is based on the discovery that part of the content from your study has already been published in another peer-reviewed journal. The specific reference we identified is: Pilbery R, Smith M, Green J, et al. PP28 An analysis of NHS 111 demand for primary care services: A retrospective cohort study. Emergency Medicine Journal 2023;40:A12. Published by BMJ in October of this year. Available from: DOI http://dx.doi.org/10.1136/emermed-2023-999.27.

Our journal's policies strictly preclude the acceptance of studies that have been previously published, in full or partially, in the peer-reviewed literature.

Additionally, your submission did not incorporate the STROBE (Strengthening the Reporting of Observational Studies in Epidemiology) checklist, an essential criterion for our review process. We fully acknowledge the time, effort, and dedication involved in research and manuscript preparation. If you decide to resubmit to our journal or another, we encourage you to address these concerns. Please feel free to contact us if you have any questions or need further details about our decision. We appreciate your interest in our journal and hope you'll consider us for future submissions. 

Reviewers' comments:

Reviewer's Responses to Questions

**Comments to the Author**

1. Is the manuscript technically sound, and do the data support the conclusions?

Reviewer #1: Yes

Reviewer #2: Yes

2. Has the statistical analysis been performed appropriately and rigorously? 

Reviewer #1: Yes

Reviewer #2: Yes

3. Have the authors made all data underlying the findings in their manuscript fully available?

Reviewer #1: No

Reviewer #2: Yes

4. Is the manuscript presented in an intelligible fashion and written in standard English?

Reviewer #1: Yes

Reviewer #2: Yes

5. Review Comments to the Author

Reviewer #1: Dear authors,

1. First of all, this study proves to be interesting and relevant in theme and scope by focusing on the NHS 111 service, vital to provide assistance for people with urgent medical care needs, optimizing the patient journey throughout Health institutions and their use of Health resources. Considering (lines 64-66) “there has been little scrutiny of the ability of primary care provision (particularly out-of-hours) to meet the demand of the NHS 111 service”, this study is of greater importance in this field. Methods supporting data analyses are well described and presented based on previous literature, allowing its replication. Furthermore, conclusions are drawn according to results extracted from collected and analyzed data. It would be interesting if future studies could analise how patient decision making explains patients (non-)adherence to their allocated 111 dispositions. Or even considering to perform this kind of data analysis in a broader area, for instance. It would also be of particular interest to discuss how results could be affected by the possibility of clinical advice being provided to call handlers without clinicians actually taking over the call themselves, thus affecting the registered data.

2. Aiming to (lines 72-74) “determine the proportion of initial healthcare contacts following the index 111 call, that were a primary care service” and to determine (lines 74-78) “what proportion of primary care service contacts were made within the specified triage timeframe, and for ED admissions, the proportion of attendances that were avoidable, stratified by time of attendance and whether an initial contact had been made with a primary care service or not”, this study relies on a retrospective cohort. A convenience sample of all 111 calls between the 1st January 2021 and 31st December 2021 was obtained and a descriptive analysis was performed. Methods supporting data analyses are clearly described and presented based on previous literature.

3. Data and sources underlying the findings described in the manuscript are not fully available. Though, the authors justify this fact by mentioning it “was derived from the Connected Yorkshire research database and as such cannot be freely shared. However, access to source data can be obtained by following the Connected Yorkshire research database application process.”.

4. In general, this manuscript is presented in an intelligible fashion and written in standard English.

By being the most up-to-dated analysis of the NHS 111 service, this article is highly beneficial to this field of study. Besides, this research article notices very import aspects such as: the influence of COVID-19 pandemic in patients attendance to primary care services; the fact that under half of the callers to 111 who were triaged to a primary care service disposition contacted a primary care service as their first post-call healthcare interaction; and the fact that around 10% of emergency department attendances met the definition of an avoidable attendance. This demonstrates the importance of taking action within populations, demystifying ideas and prejudices as well as reinforcing their health literacy levels.

I would like to mention the visually interesting and dynamic way authors presented results, namely the 111 call volume by hour and day of week (Fig 1.) or the Sankey diagram of healthcare service access by patients following index 111 call (Fig 2.). It allows a fluid comprehension of these particular data.

Given all these aspects, I recommend your manuscript for publication.

Best regards.

Reviewer #2: This is a study published in another peer-reviewed journal on October 1, 2023. Published by BMJ. Pilbery R, Smith M, Green J, et alPP28 An analysis of NHS 111 demand for primary care services: A retrospective cohort study. Emergency Medicine Journal 2023;40:A12. Available from: Doi http://dx.doi.org/10.1136/emermed-2023-999.27.

The ”PLOS ONE does not accept for publication studies that have already been published, in whole or in part, elsewhere in the peer-reviewed literature." "In addition, we will not consider submissions that are currently under consideration for publication elsewhere.” <https: criteria-for-publication="" journals.plos.org="" plosone="" s="">

The Equator Network checklist was not found.</https:>

6. PLOS authors have the option to publish the peer review history of their article (what does this mean?). If published, this will include your full peer review and any attached files.

Reviewer #1: **Yes: **Bruno Filipe Coelho da Costa

Reviewer #2: No

- - - - -

---

## [Author Response · Author response to Decision Letter 0]

28 Nov 2023

Dear Dr. Ramalho,

Please find the attached re-submission as requested following the successful appeal of the editorial decision. I have taken the opportunity to make a few grammatical changes to the text, and have uploaded a copy of the manuscript with track changes enabled.

As requested, the following is a list of author responses your comments and those of the reviewers:

1. Previously published work: Already addressed by successful appeal.

2. No inclusion of STROBE checklist. Having reviewed the uploaded files, I can see that I did include a RECORD checklist, which is one of the many extensions of STROBE and in our view, more appropriate for this study.

3. We are grateful that both reviewers found the paper to be technically sound, that the statistical analysis was performed appropriately and rigorously, and that the paper was well written.

4. We acknowledge that the source data was not able to be provided, but we have explained why this is within the data availability section, and how an academic researcher wishing to replicate the results, can apply for permission to the relevant research database to undertake this. It is my understanding that research databases providing patient-level data, typically do not allow researchers to freely share this confidential data publicly.

5. Future studies. We are grateful for reviewer’s 1 suggestion about future studies and exploration of this work. There are opportunities for qualitative and quantitative research and we will be considering these in future work.

6. We are grateful for the recognition of the visually appealing figures (Figure 1 and 2). These can be time-consuming to create but feel strongly that these help the reader navigate the rich data presented. 

We hope that these address any outstanding issues preventing publication, but as always, would be pleased to hear from you about further revision, if required.

---

## [Decision Letter · Decision Letter 1]

11 Feb 2024

PONE-D-23-09286R1An analysis of NHS 111 demand for primary care services: A retrospective cohort studyPLOS ONE

Dear Dr. Pilbery,

Thank you for submitting your manuscript to PLOS ONE. After careful consideration, we feel that it has merit but does not fully meet PLOS ONE’s publication criteria as it currently stands. Therefore, we invite you to submit a revised version of the manuscript that addresses the points raised during the review process.

We look forward to receiving your revised manuscript.

Kind regards,

Adam R. Aluisio, M.D MSc., DTM&H

Academic Editor

PLOS ONE

Journal Requirements:

2. Please note that in order to use the direct billing option the corresponding author must be affiliated with the chosen institute. Please either amend your manuscript to change the affiliation or corresponding author, or email us at plosone@plos.org with a request to remove this option

4. Please upload a Response to Reviewers letter which should include a point by point response to each of the points made by the Editor and / or Reviewers. (This should be uploaded as a 'Response to Reviewers' file type.) Please follow this link for more information: http://blogs.PLOS.org/everyone/2011/05/10/how-to-submit-your-revised-manuscript/

Additional Editor Comments (if provided):

Thank you for re-submitting your draft manuscript to PLOS One for consideration for publication. Your draft manuscript completed peer-review and the majority of concerns have been satisfied. As a minor revision please complete and provide as a supplement the RECORD checklist for your draft manuscript. Thereafter it should be acceptable for publication.

Reviewers' comments:

Reviewer's Responses to Questions

**Comments to the Author**

1. If the authors have adequately addressed your comments raised in a previous round of review and you feel that this manuscript is now acceptable for publication, you may indicate that here to bypass the “Comments to the Author” section, enter your conflict of interest statement in the “Confidential to Editor” section, and submit your "Accept" recommendation.

Reviewer #1: (No Response)

Reviewer #2: All comments have been addressed

2. Is the manuscript technically sound, and do the data support the conclusions?

Reviewer #1: Yes

Reviewer #2: Yes

3. Has the statistical analysis been performed appropriately and rigorously? 

Reviewer #1: Yes

Reviewer #2: Yes

4. Have the authors made all data underlying the findings in their manuscript fully available?

Reviewer #1: Yes

Reviewer #2: Yes

5. Is the manuscript presented in an intelligible fashion and written in standard English?

Reviewer #1: Yes

Reviewer #2: Yes

6. Review Comments to the Author

Reviewer #1: Dear authors,

According to our journal´s policies, it is precluded the acceptance of studies that have been previously published, in full or partially, in the peer-reviewed literature. This is stated on PLOS ONE publication criteria, namely “PLOS ONE does not accept for publication studies that have already been published, in whole or in part, elsewhere in the peer-reviewed literature.” ; “In addition, we will not consider submissions that are currently under consideration for publication elsewhere.”.

Part of the content from your study has already been published in another peer-reviewed journal (BMJ, October 1, 2023) under the following reference: Pilbery R, Smith M, Green J, et alPP28 An analysis of NHS 111 demand for primary care services: A retrospective cohort study. Emergency Medicine Journal 2023;40:A12. Available from: Doi http://dx.doi.org/10.1136/emermed-2023-999.27.

Subsequently, your manuscript does not meet our criteria for publication and must therefore be rejected. Thank you for your effort and dedication involved in this manuscript preparation and submission.

Reviewer #2: (No Response)

7. PLOS authors have the option to publish the peer review history of their article (what does this mean?). If published, this will include your full peer review and any attached files.

Reviewer #1: **Yes: **Bruno Filipe Coelho da Costa

Reviewer #2: **Yes: **Abel Silva de Meneses

---

## [Author Response · Author response to Decision Letter 1]

19 Feb 2024

Thank you for your recent email regarding the outcome of the resubmission peer review. The only outstanding item is your request for an upload of the RECORD checklist, which I have completed.

In addition, there were a number of editorial administrative requests:

1. Manuscript meeting PLOS ONE style requirements. We have reviewed the PLOS ONE style requirements and are confident that our manuscript meets those outlined in the linked style documents.

2. Corresponding author affiliation. I have added my University of Sheffield affiliation, in addition to my clinical one.

3. Data availability statement. We have amended the statement to match the template suggested by PLOS One.

Reviewer’s comments

4. Previous publication. This is a poster abstract publication that was identified in an earlier submission. Following an appeal, where it was confirmed that I had followed the PLOS ONE guidance, this submission was reinstated.

---

## [Editor Report · Decision Letter 2]

23 Feb 2024

An analysis of NHS 111 demand for primary care services: a retrospective cohort study

PONE-D-23-09286R2

Dear Dr. Pilbery,

We’re pleased to inform you that your manuscript has been judged scientifically suitable for publication and will be formally accepted for publication once it meets all outstanding technical requirements.

Kind regards,

Adam R. Aluisio, M.D MSc., DTM&H

Academic Editor

PLOS ONE

Additional Editor Comments (optional):

Thank you for revising.
---

## [Editor Report · Acceptance letter]

17 Apr 2024

PONE-D-23-09286R2 

PLOS ONE

Dear Dr. Pilbery, 

I'm pleased to inform you that your manuscript has been deemed suitable for publication in PLOS ONE. Congratulations! Your manuscript is now being handed over to our production team.

Kind regards, 

on behalf of

Dr. Adam R. Aluisio 

Academic Editor

PLOS ONE